# STILL NOT SYSTEMATIC AFTER ALL THESE YEARS: ON THE COMPOSITIONAL SKILLS OF SEQUENCE-TO-SEQUENCE RECURRENT NETWORKS

## ABSTRACT

Humans can understand and produce new utterances effortlessly, thanks to their systematic compositional skills. Once a person learns the meaning of a new verb "dax," he or she can immediately understand the meaning of "dax twice" or "sing and dax." In this paper, we introduce the SCAN domain, consisting of a set of simple compositional navigation commands paired with the corresponding action sequences. We then test the zero-shot generalization capabilities of a variety of recurrent neural networks (RNNs) trained on SCAN with sequence-to-sequence methods. We find that RNNs can generalize well when the differences between training and test commands are small, so that they can apply "mix-and-match" strategies to solve the task. However, when generalization requires systematic compositional skills (as in the "dax" example above), RNNs fail spectacularly. We conclude with a proof-of-concept experiment in neural machine translation, supporting the conjecture that lack of systematicity is an important factor explaining why neural networks need very large training sets.

## 1 INTRODUCTION

Human language and thought are characterized by *systematic compositionality*, the algebraic capacity to understand and produce a potentially infinite number of novel combinations from known components. For example, if a person knows the meaning and usage of words such as "twice," "and," and "again," once she learns a new verb such as "to dax" she can immediately understand or produce instructions such as "dax twice and then dax again." This type of compositionality is central to the human ability to make strong generalizations from very limited data. In a set of influential and controversial papers, Jerry Fodor and other researchers have argued that neural networks are not plausible models of the mind because they are associative devices that cannot capture systematic compositionality (Fodor & Pylyshyn, 1988; Marcus, 1998; Fodor & Lepore, 2002; Marcus, 2003; Calvo & Symons, 2014, a.o.).

In the last few years, neural network research has made astounding progress in practical domains where success crucially depends on the generalization capabilities of a system. Perhaps most strikingly, *end-to-end recurrent neural networks* currently dominate the state-of-the-art in machine translation (Bojar et al., 2016; Wu et al., 2016).[1] Since the overwhelming majority of sentences or even word sequences in a language only occur once, even in a large corpus (Baroni, 2009), this points to strong generalization abilities. Still, it is commonly observed that neural networks are extremely sample inefficient, requiring very large training sets, which suggests they may lack the same algebraic compositionality that humans exploit, and they might only be sensitive to broad patterns over lots of accumulated statistics (Lake et al., 2016).

In this paper, we introduce a grounded navigation environment where the learner must translate commands given in a limited form of natural language into a sequence of actions. This problem is naturally framed as a sequence-to-sequence task, and, due to its simplicity, it is ideal to study the systematic generalization capabilities of computational systems to novel examples in a controlled

---

[1]Very recently, *convolutional* networks have reached comparable or superior performance on the same task (Gehring et al., 2017). We leave the investigation of their systematicity to future work.

| | | |
|---|---|---|
| jump | ⇒ | JUMP |
| jump left | ⇒ | LTURN JUMP |
| jump around right | ⇒ | RTURN JUMP RTURN JUMP RTURN JUMP RTURN JUMP |
| turn left twice | ⇒ | LTURN LTURN |
| jump thrice | ⇒ | JUMP JUMP JUMP |
| jump opposite left and walk thrice | ⇒ | LTURN LTURN JUMP WALK WALK WALK |
| jump opposite left after walk around left | ⇒ | LTURN WALK LTURN WALK LTURN WALK LTURN WALK LTURN LTURN JUMP |

Figure 1: Examples of SCAN commands (left) and the corresponding action sequences (right).

setup. We thus use it to test a wide range of modern recurrent network architectures in terms of their compositional skills. Our results suggest that, although standard architectures such as LSTMs with attention (Bahdanau et al., 2015) do generalize when novel examples feature a mixture of constructions that have been observed in training, the models are catastrophically affected by *systematic* differences between training and test sentences, of the sort that would be trivial for an agent equipped with an "algebraic mind" (Marcus, 2003).

## 2 THE SCAN TASKS

We call our data set SCAN because it is a **S**implified version of the **C**omm**AI** **N**avigation tasks (Mikolov et al., 2016).[2] For a learner, the goal is to translate commands presented in simplified natural language into a sequence of actions. Since each command is unambiguously associated to a single action sequence, SCAN (unlike the original CommAI tasks) can be straightforwardly treated as a supervised sequence-to-sequence semantic parsing task (Dong & Lapata, 2016; Jia & Liang, 2016; Herzig & Berant, 2017), where the input vocabulary is given by the set of words used in the commands, and the output by the set of actions available to the learner.

Several examples from SCAN are presented in Fig. 1. Formally, SCAN consists of all the commands generated by a phrase-structure grammar (see Appendix Fig. 6) and the corresponding sequence of actions, produced according to a semantic interpretation function (Appendix Fig. 7). Intuitively, the SCAN grammar licenses commands denoting primitive actions such as JUMP (denoted by "jump"; Fig. 1), WALK (denoted by "walk") and LTURN (denoted by "turn left"). We will refer to these as *primitive commands*. It also accepts a set of modifiers and conjunctions that compositionally build expressions referring to action sequences. The "left" and "right" modifiers take commands referring to undirected primitive actions as input and return commands denoting their directed counterparts ("jump left"; Fig. 1). The "opposite" modifier produces an action sequence that turns the agent backward in the specified direction before executing a target action ("jump opposite left"), while "around" makes the agent execute the action at each step while turning around in the specified direction ("jump around right"; Fig. 1). The "twice/thrice" modifiers trigger repetition of the command they take scope over, and "and/after" combine two action sequences. Although the SCAN examples in Fig. 1 focus on the "jump"/JUMP primitive, each instance of JUMP can be replaced with either WALK, RUN, or LOOK to generate yet more commands. Many more combinations are possible as licensed by the grammar.

The SCAN grammar, lacking recursion, generates a finite but large set of commands (20,910, to be precise). Commands can be decoded compositionally by applying the function in Appendix Fig. 7. This means that, if it acquires the right interpretation function, a learner can understand commands it has not seen during training. For example, the learner might have only observed the primitive "jump" command during training, but if it has learned the meaning of "after", "twice" and "around left" from other verbs, it should be able to decode, zero-shot, the complex command: "jump around left after jump twice".

## 3 MODELS AND SETUP

We approach SCAN through the popular sequence-to-sequence (seq2seq) framework, in which two recurrent networks work together to learn a mapping between input sequences and output sequences

---

[2]SCAN available at: `http://anonymized.com`

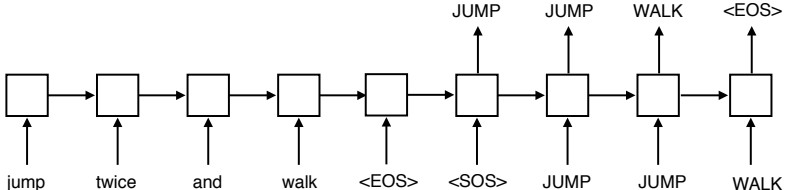

Figure 2: How the seq2seq framework is applied to SCAN. The symbols <EOS> and <SOS> denote end-of-sentence and start-of-sentence, respectively. The encoder (left) ends with the first <EOS> symbol, and the decoder (right) begins with the <SOS> symbol.

(e.g., Sutskever et al., 2014). Fig. 2 illustrates the application of the seq2seq approach to a SCAN example. First, a recurrent network encoder receives the input sequence word-by-word, forming a low-dimensional representation of the entire command. Second, the low-dimensional representation is passed to a recurrent network decoder, which then generates the output sequence action-by-action. The decoder's output is compared with the ground truth, and the backpropagation algorithm is used to the update the parameters of both the encoder and decoder. Note that although the encoder and decoder share the same network structure (e.g., number of layers and hidden units), they do not otherwise share weights/parameters with each other. More details regarding the encoder-decoder RNN are provided in the Appendix.

Using the seq2seq framework, we tested a range of standard recurrent neural network models from the literature: simple recurrent networks (SRNs; Elman, 1990), long short-term memory networks (LSTMs; Hochreiter & Schmidhuber, 1997), and gated recurrent units (GRUs; Chung et al., 2014). Recurrent networks with attention have become increasingly popular in the last few years, and thus we also tested each network with and without an attentional mechanism, using the attentional model from Bahdanau et al. (2015) (see Appendix for more details). Finally, to make the evaluations as systematic as possible, a large-scale hyperparameter search was conducted that varied the number of layers (1 or 2), the number of hidden units per layer (25, 50, 100, 200, or 400), and the amount of dropout (0, 0.1, 0.5; applied to recurrent layers and word embeddings). Varying these hyperparameters leads to 180 different network architectures, all of which were run on each experiment and replicated 5 times each with different random initializations.[3]

In reporting the results and analyzing the successes and failures of the networks, we focus on the **overall-best** architecture as determined by the extensive hyperparameter search. The winning architecture was a **2-layer LSTM with 200 hidden units per layer, no attention, and dropout applied at the 0.5 level**. Although the detailed analyses to follow focus on this particular architecture, the top-performing architecture for each experiment individually is also reported and analyzed.

Networks were trained with the following specifications. Training consisted of 100,000 trials, each presenting an input/output sequence and then updating the networks weights. The ADAM optimization algorithm was used with default parameters, including a learning rate of 0.001 (Kingma & Welling, 2014). Gradients with a norm larger than 5.0 were clipped. Finally, the decoder requires the previous step's output as the next step's input, which was computed in two different ways. During training, for half the time, the network's self-produced outputs were passed back to the next step, and for the other half of the time, the ground-truth outputs were passed back to the next step (teacher forcing; Williams & Zipser, 1989). The networks were implemented in PyTorch and based on a standard seq2seq implementation.[4]

## 4 EXPERIMENTS

In each of the following experiments, the recurrent networks are trained on a large set of commands from the SCAN tasks to establish background knowledge. The networks were successful at mastering the background tasks: Training accuracy was above 99.5% for the overall-best network in each of the key experiments, and it was at least 95% or above for the top-performers in each experiment

---

[3]A small number of runs (23/3600) did not complete, and thus not every network had 5 runs.

[4]http://pytorch.org/tutorials/intermediate/seq2seq_translation_tutorial.html

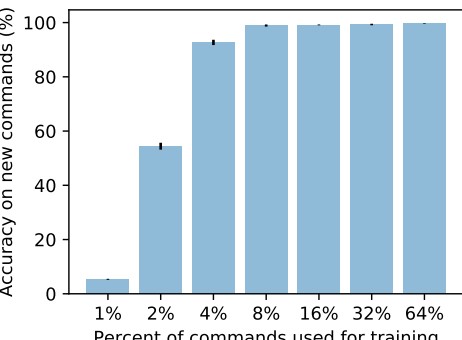

Figure 3: Zero-shot generalization after training on a random subset of the SCAN tasks. The overall-best network was trained on varying proportions of the corpus (x-axis) and generalization was measured on new tasks (y-axis). Each bar shows the mean over 5 training runs with corresponding ±1 SEM. Note that the SCAN tasks were randomly split once, and progressively more examples were added to form the more varied training subsets.

specifically. After training, the networks are then evaluated on new commands designed to test generalization beyond the background set in systematic, compositional ways. In evaluating these new commands, the networks must make zero-shot generalizations and produce the appropriate action sequence based solely on extrapolation from the background training.

EXPERIMENT 1: GENERALIZING TO A RANDOM SUBSET OF COMMANDS

In this straightforward experiment, the SCAN tasks were randomly split into a training set (80%) and a test set (20%). The training set provides broad coverage of the task space, and the test set examines how networks can decompose and recombine commands from the training set. For instance, the network is asked to perform the new command, "**jump opposite right after walk around right thrice**," as a zero-shot generalization in the test set. Although the conjunction as a whole is novel, the parts are not: The training set features many examples of the parts in other contexts, e.g., "**jump opposite right after** turn opposite right" and "jump right twice **after walk around right thrice**"(both bold sub-strings appear 83 times in the training set). To succeed, the network needs to make compositional generalizations, recombining pieces of existing commands to perform new ones.

Overall, the networks were highly successful at generalizing to random SCAN commands. The top-performing network for this experiment achieved 99.8% correct on the test set (accuracy values here and below are averaged over five training runs). The top-performing architecture was a LSTM with no attention, 2 layers of 200 hidden units, and no dropout. The best-overall network achieved 99.7% percent correct. Interestingly, not every architecture was successful: Classic SRNs performed very poorly, and the best SRN achieved less than 1.5% correct at test time (performance on the training set was equally low). However, attention-augmented SRNs learned the commands much better, achieving 59.7% correct on average for the test set (with a range between 18.4% and 94.0% across SRN architectures). For LSTMs and GRUs, attention was not essential, because many of the highest performing architectures did not use it.

It is not yet clear how much background knowledge is required to learn the underlying compositional structure of the tasks. As indicated above, the main split was quite generous, providing 80% of the commands at training time for a total of over 16,700 distinct examples (with strong combinatorial coverage). We next re-trained the best-overall network with varying numbers of distinct examples. The results are shown in Fig. 3. With 1% of the commands shown during training (about 210 examples), the network performs poorly at about 5% correct. With 2% coverage, performance improves to about 54% correct on the test set. By 4% coverage, performance is about 93% correct. Our results show that not only can networks generalize to random subsets of the tasks, they can do so from relatively sparse coverage of the compositional command space. Still, even with this sparser coverage, differences between training and test instances are not dramatic. Let's for example consider the set of all commands without a conjunction (e.g., "walk around thrice", "run", "jump opposite left twice"). All the commands of this sort that occur in the test set of the 2% training coverage split (either as components of a conjunction or by themselves) *also* occur in the corresponding training set, with an average of 8 occurrences. Even for the 1% split, there is only one conjunction-less test command that does not also occur in the training split, and the average frequency of occurrence of such commands in the training set is at a non-negligible value of 4 times.

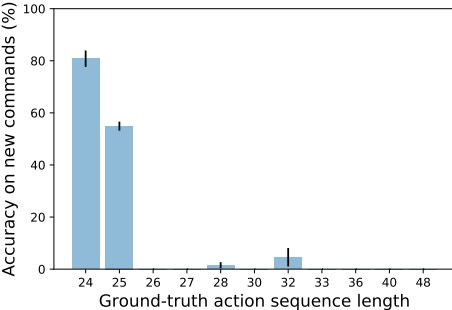 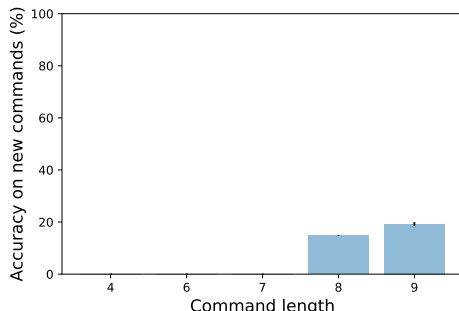

Figure 4: Zero-shot generalization to commands with action sequence lengths not seen in training. Left: accuracy distribution by action sequence length; right: accuracy distribution by command length (only lengths attested in the test set shown, in both cases). Bars show means over 5 runs of overall-best model with $\pm 1$ SEM.

EXPERIMENT 2: GENERALIZING TO COMMANDS DEMANDING LONGER ACTION SEQUENCES

The previous experiment confirmed that sequence-to-sequence RNNs can zero-shot generalize to new commands. This is not too surprising, as otherwise they could not have achieved the impressive results they reached in machine translation and other domains. However, the random split we considered above implies that the degree of generalization required to understand the test commands and to produce action sequences will be generally minor, and could be performed by recombining pieces of seen commands/action sequences within familiar templates, as we discussed.

We study next a more *systematic* form of generalization, where models must bootstrap to commands requiring longer action sequences than those seen in training.[5] Now the training set contains all 16,990 commands requiring sequences of up to 22 actions, whereas the test set includes all remaining commands (3,920, requiring action sequences of lengths from 24 to 48). Under this split, for example, at test time the network must execute the command "jump around left twice and walk opposite right thrice", requiring a sequence of 25 actions. While all the elements used in the command have been observed during training, the network has never been asked to produce a sequence of this length, nor it has ever seen an "around * twice" command conjoined with an "opposite * thrice" command (although it did observe both components conjoined with others). Thus, it must productively generalize familiar verbs, modifiers and conjunctions to generate longer action sequences.

This test turns out to be very challenging for all models. The best result (20.8% on average, again over 5 runs) is achieved by a GRU with attention, one 50-dimensional hidden layer, and dropout 0.5 (interestingly, a model with considerably less capacity than the best for the random-split setup). The overall-best model achieves 13.8% accuracy.

Fig. 4 (left) shows partial success is almost entirely explained by generalization to the shortest action sequence lengths in the test set. The right panel of Fig. 4 shows accuracy in the test set organized by command length (in word tokens). The model only gets right some of the *longest* commands (8 or 9 tokens). In the training set, the longest action sequences ($\geq 20$) are invariably associated to commands containing 8 or 9 tokens. Thus, the model is correctly generalizing only in those cases that are most similar to training instances.

Finally, we studied whether the difficulty with long sequences can be mitigated if the proper length was provided by an oracle at evaluation time.[6] If this difficulty is a relatively straightforward issue of the decoder terminating too early, then this should provide an (unrealistic) fix. If this difficulty is symptomatic of deeper problems with generalization, then this change will have only a small effect. With the oracle, the overall-best network performance improved from 13.8% to 23.6% correct, which was notable but insufficient to master the long sequences. The top-performing model showed a more substantial improvement (20.8% to 60.2%). Although improved, the networks were far from

---

[5]We focus on action sequence length rather than command length since the former exhibits more variance (1-48 vs. 1-9).

[6]Any attempt from the decoder to terminate the action sequence with an <EOS> was ignored (and the second strongest action was chosen) until a sequence with proper length was produced.

| run | | jump | | run twice | | jump twice | |
|---|---|---|---|---|---|---|---|
| look | .73 | *run* | *.15* | look twice | .72 | *walk and walk* | *.19* |
| walk | .65 | *walk* | *.13* | run twice and look opposite right thrice | .65 | *run and walk* | *.16* |
| walk after run | .55 | *turn right* | *.12* | run twice and run right twice | .64 | *walk opposite right and walk* | *.12* |
| run thrice after run | .50 | *look right twice after walk twice* | *.09* | run twice and look opposite right twice | .63 | *look right and walk* | *.12* |
| run twice after run | .49 | *turn right after turn right* | *.09* | walk twice and run twice | .63 | *walk right and walk* | *.11* |

Table 1: Nearest training-element hidden representations for a sample of commands, with the respective cosines. Here, "jump" was trained in isolation while "run" was trained compositionally. Italics are used to emphasize large-distance items (cosine <0.2).

perfect and still exhibited key difficulties with long sequences of output actions (again, even for the top model, there was a strong effect of action sequence length, with average accuracy ranging from 95.76% for commands requiring 24 actions to 22.8% for commands requiring 48 actions).

EXPERIMENT 3: GENERALIZING COMPOSITION ACROSS PRIMITIVE COMMANDS

Our next experiment comes closest to testing Fodor's view of systematicity. In the training phase, the model is exposed to the primitive command only denoting a certain basic action (e.g., "jump"). The model is also exposed to all primitive and composed commands for all other actions (e.g., "run", "run twice", "walk", "walk opposite left and run twice", etc.). At test time, the model has to execute all composed commands for the action that it only saw in the primitive context (e.g., "jump twice", "jump opposite left and run twice", etc.). According to the classic thought experiments of Fodor and colleagues, this should be easy: if you know the meaning of "run", "jump" and "run twice", you should also understand what "jump twice" means.

We run two variants of the experiment generalizing from "turn left" and "jump", respectively. Since "turn right" is distributionally identical to "turn left" and "walk", "run" and "look" are distributionally identical to "jump", it is redundant to test all commands. Moreover, to ensure the networks were highly familiar with the target primitive command ("jump" or "turn left"), it was over-represented in training such that roughly 10% of all training presentations were of the command.

We obtain strikingly different results for "turn left" and "jump". For "turn left", many models generalize very well to composed commands. The best performance is achieved by a GRU network with attention, one layer with 100 hidden units, and dropout of 0.1 (90.3% accuracy). The overall-best model achieved 90.0% accuracy. On the other hand, for "jump," models are almost completely incapable to generalize to composed commands. The best performance was 1.2% accuracy (LSTM, attention, one layer, 100 hidden units, dropout 0.1). The overall-best model reached 0.08% accuracy.

In the case of "turn left", although models are only exposed to the primitive command during training, they will see the action it denotes (LTURN) many times, as it is used to accomplish many directed actions. For example, a training example is: "walk left and jump left", with ground-truth interpretation: LTURN WALK LTURN JUMP. Apparently, seeing *action sequences* containing LTURN suffices for the model to understand composed *commands* with "turn left". On the other hand, the action denoted by "jump" (JUMP) only occurs with this primitive command in training, and the model does not generalize from this minimal context to new composed ones.

Looking at the results in more details (for the median-performance run of the overall-best model), we observe that even in the successful "turn left" case the model errors are surprising. One would expect such errors to be randomly distributed, or perhaps to pertain to the longest commands or action sequences. Instead, all 45 errors made by the model are conjunctions where one of the components is simple "turn left" (22 cases) or "turn left thrice" (23 cases). This is particularly striking because the network produced the correct mapping for "turn left" during training, as well as for "turn left thrice" at test time, and it gets many more conjunctions right (ironically, including "turn left thrice and turn left", "turn left thrice after turn left" etc.). We conclude that, even when the network has apparently learned systematic composition almost perfectly, it got at it in a very non-human-like way: it's hard to conceive of someone who understood the meaning of "turn left", and "jump right

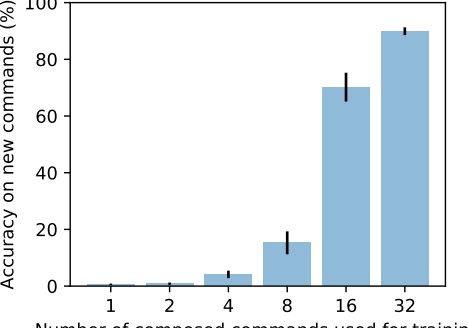

Figure 5: Zero-shot generalization after adding the primitive "jump" and some compositional jump commands. The overall-best network was trained on different numbers of composed "jump" commands (x-axis), and generalization was measured on new composed "jump" commands (y-axis). Each bar shows the mean over 5 runs with varying training commands along with the corresponding $\pm 1$ SEM.

and turn left twice" (which the network gets right), but not that of "jump right and turn left" (one of the examples the network missed). In the "jump" experiment, the network could only correctly decode two composite cases, both starting with the execution of primitive "jump", conjoined with a different action: "jump and run opposite right", "jump and walk around left thrice".

It is instructive to look at the representations that the network induced for various commands in the latter experiment. Table 1 reports the 5 nearest neighbours for a sample of commands. Command similarity is measured by the cosine between the final decoder hidden state vectors, and computed with respect to all commands present in the training set. "Run" is provided as an example primitive command for which the model has been exposed to the full composed paradigm in training. As one would expect, "run" is close to the other primitive commands ("look", "walk"), as well as to short conjoined commands that contain primitive "run" as one of the conjuncts (we observe a similar pattern for the "jump" representation induced in Experiment 1). Instead, since "jump" had a different training distribution than the other primitive commands, the model does not capture its similarity to them, as shown by the very low cosines of its nearest "neighbours". Since it fails to establish a link to other basic commands, the model does not generalize modifier application from them to "jump". Although "run twice" is similar to (conjunctions of) other primitive tasks composed with "twice", "jump twice" is isolated in representational space, and its (far) nearest neighbours look arbitrary.

We tested here systematicity in its purest form: the model was only exposed to "jump" in isolation, and asked to bootstrap to its compositional paradigm based on the behaviour of other primitive commands such as "walk", "look" and "run". Although we suspect humans would not have problems with this setup, it arguably is too opaque for a computational model, which could lack evidence for "jumping" being the same sort of action as "walking". Suppose we give the network *some* evidence that "jumping" composes like "walking" by showing a few composed "jump" command during training. Is the network then able to generalize to the full composed paradigm?

This question is answered in Figure 5. Again, the new primitive command (and its compositions) were over-sampled during training to make up 10% of all presentations. Here, even when shown 4 different composed commands with "jump", the network does not generalize to other composed commands (4.1% correct). Weak generalization starts appearing when the network is presented 8 composed tasks in training (15.3%), and significant generalization (still far from perfect) shows up when the training set contains 16 and especially 32 distinct composed commands (70.2% and 89.9%, respectively). We conclude that the network is not failing to generalize simply because, in the original setup, it had no evidence that "jump" should behave like the other commands. On the other hand, the runs with more composed examples confirm that, as we found in Experiment 1, the network does display powerful generalization abilities. Simply, they do not conform to the "all-or-nothing" rule-based behaviour we would expect from a systematically compositional device–and, as a consequence, they require more positive examples to emerge.

### EXPERIMENT 4: COMPOSITIONALITY IN MACHINE TRANSLATION

Our final experiment is a proof-of-concept that our findings are more broadly applicable; that is, the limitations of recurrent networks with regards to systematic compositionality extend beyond SCAN to other sequence-to-sequence problems such as machine translation. First, to test our setup for machine translation, we trained our standard seq2seq code on short ($\leq$ 9 words) English-French

sentence pairs that begin with English phrases such as "I am," "he is," "they are," and their contractions (randomly split with 10,000 for training and 1180 for testing).[4] An informal hyperparameter search led us to pick a LSTM with attention, 2 layers of 400 hidden units, and 0.05 dropout. With these hyperparameters and the same training procedure used for the SCAN tasks (Section 3), the network reached a respectable 28.6 BLEU test score after 100,000 steps.

Second, to examine compositionality with the introduction of a new word, we trained a fresh network after adding 1,000 repetitions of the sentence "I am daxy" (fr. "je suis daxiste") to the training data (the BLEU score on the original test set dropped less than 1 point). We tested this network by embedding "daxy" into the following constructions: "you are daxy" ("tu es daxiste"), "he is daxy" ("il est daxiste"), "I am not daxy" ("je ne suis pas daxiste"), "you are not daxy" ("tu n'es pas daxiste"), "he is not daxy" ("il n'est pas daxiste"), "I am very daxy" ("je suis très daxiste"), "you are very daxy" ("tu es très daxiste"), "he is very daxy" ("il est très daxiste"). During training, the model saw these constructions occurring with 22 distinct predicates on average (limiting the counts to perfect matches, excluding, e.g., "you are not very X"). Still, the model could only get one of the 8 translations right (that of "he is daxy"). For comparison, for the adjective "tired", which occurred in 80 different constructions in the training corpus, our model had 8/8 accuracy when testing on the same constructions as for "daxy" (only one of which also occurred with "tired" in the training set). Although this is a small-scale machine translation problem, our preliminary result suggests that models will similarly struggle with systematic compositionality in larger data sets, when adding a new word to their vocabulary, in ways that people clearly do not.

## 5    DISCUSSION

In the thirty years since the inception of the systematicity debate, many authors on both sides have tested the ability of neural networks to solve tasks requiring compositional generalization, with mixed results (e.g., Christiansen & Chater, 1994; Marcus, 1998; Phillips, 1998; Chang, 2002; van der Velde et al., 2004; Wong & Wang, 2007; Brakel & Frank, 2009; Frank et al., 2009; Frank, 2014). However, to the best of our knowledge, ours is the first study testing systematicity in modern seq2seq models, and our results confirm the mixed picture. On the one hand, standard recurrent models can reach very high zero-shot accuracy from relatively few training examples, as long as the latter are generally representative of the test data (Experiment 1). However, the same networks fail spectacularly when there are systematic differences between training and testing. Crucially, the training data of the relevant experiments provide enough evidence to learn composition rules affording the correct generalizations. In Experiment 2, the training data contain examples of all modifiers and connectives that are needed at test time for producing longer action sequences. In Experiment 3, the usage of modifiers and connectives is illustrated at training time by their application to some primitive commands, and, at test time, the model should apply them to a new command it encountered in isolation during training. Nonetheless, this evidence was not sufficient for each of the networks we tested. Generalization only occurs when the networks are also exposed to the target command (or the corresponding action) in a fair number of composed contexts during training.

Given the astounding successes of seq2seq models in challenging tasks such as machine translation, one might argue that failure to generalize by systematic composition indicates that neural networks are poor models of some aspects of human cognition, but it is of little practical import. However, systematicity is an extremely efficient way to generalize. Once a person learns the new English adjective "daxy", he or she can immediately produce and understand an infinity of sentences containing it. The SCAN experiments and a proof-of-concept machine translation experiment (Experiment 4) suggest that this ability is still beyond the grasp of state-of-the-art networks, likely contributing to their striking sample-inefficiency. These results give us hope that a model capable of systematic compositionality could greatly benefit machine translation and other applications.

A natural way of achieving stronger compositionality is through learning more structured representations. Recently, neural networks with external memories have shown promise for extracting algorithm-like representations from input/output examples (Joulin & Mikolov, 2015; Graves et al., 2016); for instance, these networks can outperform standard RNNs on generalizing to longer sequences. Future work will explore these approaches on SCAN and other tests of zero-shot compositional generalization. Ultimately, we see systematic compositionality as key both to developing more powerful algorithms and to enriching our computational understanding of the human mind.

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

APPENDIX

| | | |
|---|---|---|
| C → S and S | V → D[1] opposite D[2] | D → turn left |
| C → S after S | V → D[1] around D[2] | D → turn right |
| C → S | V → D | U → walk |
| S → V twice | V → U | U → look |
| S → V thrice | D → U left | U → run |
| S → V | D → U right | U → jump |

Figure 6: Phrase-structure grammar generating SCAN commands. We use indexing notation to allow infixing: D[i] is to be read as the i-th element directly dominated by category D.

⟦walk ⟧ = WALK
⟦look⟧ = LOOK
⟦run⟧ = RUN
⟦jump⟧ = JUMP
⟦turn left⟧ = LTURN
⟦turn right⟧ = RTURN
⟦u left⟧ = LTURN ⟦u⟧
⟦u right⟧ = RTURN ⟦u⟧
⟦turn opposite left⟧ = LTURN LTURN
⟦turn opposite right⟧ = RTURN RTURN
⟦u opposite left⟧ = ⟦turn opposite left⟧ ⟦u⟧

⟦u opposite right⟧ = ⟦turn opposite right⟧ ⟦u⟧
⟦turn around left⟧ = LTURN LTURN LTURN LTURN
⟦turn around right⟧ = RTURN RTURN RTURN RTURN
⟦u around left⟧ = LTURN ⟦u⟧ LTURN ⟦u⟧ LTURN ⟦u⟧ LTURN ⟦u⟧
⟦u around right⟧ = RTURN ⟦u⟧ RTURN ⟦u⟧ RTURN ⟦u⟧ RTURN ⟦u⟧
⟦x twice⟧ = ⟦x⟧ ⟦x⟧
⟦x thrice⟧ = ⟦x⟧ ⟦x⟧ ⟦x⟧
⟦x₁ and x₂⟧ = ⟦x₁⟧ ⟦x₂⟧
⟦x₁ after x₂⟧ = ⟦x₂⟧ ⟦x₁⟧

Figure 7: Double brackets (⟦⟧) denote the interpretation function translating SCAN's linguistic commands into sequences of actions (denoted by uppercase strings). Symbols $x$ and $u$ denote variables, the latter limited to words in the set {walk, look, run, jump}. The linear order of actions denotes their temporal sequence.

STANDARD ENCODER-DECODER RNN

In this section, we describe the encoder-decoder framework, borrowing from the description in Bahdanau et al. (2015). The encoder receives a natural language command as a sequence of $T$ words. The words are transformed into a sequence of vectors, $\{w_1, \ldots, w_T\}$, which are learned embeddings with the same number of dimensions as the hidden layer. A recurrent neural network (RNN) processes each word

$$h_t = f_E(h_{t-1}, w_t),\tag{1}$$

where $h_t$ is the encoder hidden state. The final hidden state $h_T$ (which may include multiple layers for multi-layer RNNs) is passed to the RNN decoder as hidden state $g_0$ (see Figure 2). Then, the RNN decoder must generate a sequence of output actions $a_1, \ldots, a_R$. To do so, it computes

$$g_t = f_D(g_{t-1}, a_{t-1}),\tag{2}$$

where $g_t$ is the decoder hidden state and $a_{t-1}$ is the (embedded) output action from the previous time step. Last, the hidden state $g_t$ is mapped to a softmax to select the next action $a_t$ from all possible actions.

ATTENTION ENCODER-DECODER RNN

For the encoder-decoder with attention, the encoder is identical to the one described above. Unlike the standard decoder that can only see $h_T$, the attention decoder can access all of the encoder hidden states, $h_1, \ldots, h_T$ (in this case, only the last layer if multi-layer). At each step $i$, a context vector $c_i$ is computed as a weighted sum of the encoder hidden states

$$c_i = \sum_{t=1}^{T} \alpha_{it} h_t.\tag{3}$$

The weights $\alpha_{it}$ are computed using a softmax function $\alpha_{it} = \exp(e_{it}) / \sum_{j=1}^{T} \exp(e_{ij})$, where $e_{it} = v_a^\top \tanh(W_a g_{i-1} + U_a h_t)$ is an alignment model that computes the similarity between the previous decoder hidden state $g_{i-1}$ and an encoder hidden state $h_t$ (for the other variables, $v_a$, $W_a$,

and $U_a$ are learnable parameters) (Bahdanau et al., 2015). This context vector $c_i$ is then passed as input to the decoder RNN at each step with the function

$$g_i = f_D(g_{i-1}, a_{i-1}, c_i), \tag{4}$$

which also starts with hidden state $g_0 = h_T$, as in the standard decoder. Last, the hidden state $g_i$ is concatenated with $c_i$ and mapped to a softmax to select new action $a_i$.

