# OpenReview forum: "Still not systematic after all these years: On the compositional skills of sequence-to-sequence recurrent networks"
_ICLR.cc/2018/Conference — Invite to Workshop Track_

### Official Review · AnonReviewer3 · 2017-11-26
**Nicely written paper describing an interesting series of experiments.**

**Rating:** 6
**Confidence:** 3

**Review:**

This paper focuses on the zero-shot learning compositional capabilities of modern sequence-to-sequence RNNs.  Through a series of experiments and a newly defined dataset, it exposes the short-comings of current seq2seq RNN architectures.  The proposed dataset, called the SCAN dataset, is a selected subset of the CommonAI navigation tasks data set.  This subset is chosen such that each command sequence corresponds to exactly one target action sequence, making it possible to apply standard seq2seq methods.  Existing methods are then compared based on how accurately they can produce the target action sequence based on the command input sequence.

The introduction covers relevant literature and nicely describes the motivation for later experiments. Description of the model architecture is largely done in the appendix, this puts the focus of the paper on the experimental section. This choice seems to be appropriate, since standard methods are used. Figure 2 is sufficient to illustrate the model to readers familiar with the literature.

The experimental part establishes a baseline using standard seq2seq models on the new dataset, by exploring large variations of model architectures and a large part of the hyper-parameter space.   This papers experimentation sections sets a positive example by exploring a comparatively large space of standard model architectures on the problem it proposes. This search enables the authors to come to convincing conclusions regarding the shortcomings of current models. The paper explores in particular:
1.) Model generalization to unknown data similar to the training set.
2.) Model generalization to data-sequences longer than the training set.
3.) Generalization to composite commands, where  a part of the command is never observed in sequence in the training set.
4.) A recreation of a similar problem in the machine translation context.
These experiments show that modern sequence to sequence models do not solve the systematicity problem, while making clear by application to machine translation, why such a solution would be desirable. The SCAN data-set has the potential to become an interesting test-case for future research in this direction.

The experimental results shown in this paper are clearly compelling in exposing the weaknesses of current seq2seq RNN models.  However, where the paper falls a bit short is in the discussion / outlook in terms of suggestions of how one can go about tackling these shortcomings.

---

> ### Author Response · Authors · 2017-12-07
> **Authors' reply to all reviewers**
>
> We thank the reviewers for very constructive feedback. We will thoroughly incorporate your suggestions in our revisions. We refer here to reviews by the name posted with the review (“AnonReviewer1”, etc.) rather than OpenReview order. Thus, R1 refers to “AnonReviewer1”.
>
> First, we would like to remark that our paper is not just about shortcomings of seq2seq models, but also about their impressive generalization strengths. Experiment 1 shows strong generalization to zero-shot cases, even when the training data covers a small fraction of task space (e.g., 8%). Even in the most challenging Experiment 3, the best models do generalize, to a certain extent, to composed usages of some primitive commands with familiar output actions ("turn left"). These are positive results (in line with the empirical achievements of seq2seq models), and in our revisions we will emphasize them more.
>
> In other cases, there were dramatic generalization failures. We see these cases as challenges, encouraging researchers to design new models that can more successfully address compositional learning. We are already seeing a positive impact from the public release of SCAN: We were contacted by several teams actively working on our challenges, and we are very excited to see the new ideas the SCAN tasks will stimulate.
>
> We were pleased that the paper was well received by all reviewers, and there was substantial agreement on its strengths (introducing an interesting challenge, detailed experimentation, careful hyperparameter search, clarity, etc.). The most significant critique, raised by R2 and R3, was that they would like more discussion of how the shortcomings of current seq2seq models can be tackled. In response to this, we will substantially expand our discussion of promising ways of addressing these shortcomings. Our current thinking on how to tackle these problems is outlined below.
>
> We believe that the crucial component that current models are missing is the ability to extract systematic rules from the training data. R2 observes that some of our experiments violate the basic assumption that training and test data should come from the same distribution. We appreciate this point, and we believe it also depends on the degree of abstraction that a model performs on the input data. A model operating in "rule space" could extract translation rules such as:
>
> translate(x and y) -> translate(x) translate(y)
> translate(x twice) -> translate(x) translate(x)
>
> Then, if the meaning of a new command, translate(“jump”), is learned at training time and acts as a variable the rules can be applied to, no further learning is needed at test time. When represented in this more abstract way, the training and test distributions are quite similar, even if they differ in terms of shallower statistics such as word frequency. We conjecture that humans generalize in this way when learning novel compositional systems, and we are currently designing behavioral experiments to verify this hypothesis.
>
> How can we encourage a general seq2seq model to extract rules from data, rather than shallower generalizations? We are considering several possibilities:
>
> 1) Learning to learn: exposing a model to a number of different environments regulated by similar rules; an objective function requiring successful generalization to new environments would force models to learn the shared general rules;
>
> 2) More structure/stronger priors: models akin to recent neural program induction or related could provide RNNs with access to a set of manually-encoded or (ideally) learned functions; the RNN job would then be to learn to compose these functions as appropriate;
>
> 3) Differentiable data structures: extending recent work on Memory Networks, Neural Turing Machines and related formalisms, a seq2seq model could be equipped with quasi-discrete memory structures, enabling separate storage of variables, which in turn might encourage abstract rule learning.
>
> R1 proposal of a model with latent stochastic variables is also interesting, and we will further explore it.
>
> Other ideas might work specifically for the SCAN tasks (e.g., ad hoc "copying" mechanisms, or special ways to induce new word embeddings). The research lines described above point instead to more general models, combining the effectiveness of current seq2seq models with more human-like generalization capabilities. In our revisions, we will include a broader discussion of the implications of our results, and these new ideas for addressing the SCAN tasks.
>
> Given our substantial positive results in some cases, and strong interest from other teams in tackling the SCAN tasks, there is significant opportunity to make progress on compositional learning and advance the state-of-the-art in seq2seq models. Our paper offers a set of concrete tasks for catalyzing this progress. We hope that the discussion phase can resolve some of the questions regarding their significance, and how to best make progress in compositional learning.

---

### Official Review · AnonReviewer1 · 2017-11-27
**Clearly written paper with informative results**

**Rating:** 7
**Confidence:** 4

**Review:**

The paper analyzed the composition abilities of Recurrent Neural Networks.  The authors analyzed the the generalization for the following scenarios

- the generalization ability of RNNs on random subset of SCAN commands
- the generalization ability of RNNs on longer SCAN commands
- The generalization ability of composition over primitive commands.

The experiments supported the hypothesis that the RNNs are able to

- generalize zero-shot to new commands.
- difficulty generalizing to longer sequence (compared to training sequences) of commands.
- the ability of the model generalizing to composition of primitive commands seem to depend heavily on the whether the action is seen during training. The model does not seem to generalize to completely new action and commands (like Jump), however, seems to generalize much better for Turn Left, since it has seen the action during training (even though not the particular commands)

Overall, the paper is well written and easy to follow. The experiments are complete. The results and analysis are informative.

As for future work, I think an interesting direction would also be to investigate the composition abilities for RNNs with latent (stochastic) variables. For example, analyzing whether the latent stochastic variables may shown to actually help with generalization of composition of primitive commands.

---

> ### Author Response · Authors · 2017-12-07
> **Thanks and please see below**
>
> Thanks for your supportive review. We reply to all reviewers jointly in our first comment to the third reviewer below.

---

### Official Review · AnonReviewer2 · 2017-11-28
**No Title**

**Rating:** 6
**Confidence:** 5

**Review:**

This paper argues about limitations of RNNs to learn models than exhibit a human-like compositional operation that facilitates generalization to unseen data, ex. zero-shot or one-shot applications. The paper does not present a new method, it only focuses on analyzing learning situations that illustrate their main ideas. To do this, they introduce a new dataset that facilitates the analysis of a Seq2Seq learning case. They conduct a complete experimentation, testing different popular RNN architectures, as well as parameter and hyperparameters values.

The main idea in the paper is that RNNs applied to Seq2Seq case are learning a representation based only on "memorizing" a mixture of constructions that have been observed during training, therefore, they can not show the compositional learning abilities exhibit by humans (that authors refer as systematic compositionality). Authors present a set of experiments designed to support this observation.

While the experiments are compelling, as I explain below, I believe there is an underlying assumption that is not considered. Performance on training set by the best model is close to perfect (99.5%), so the model is really learning the task. Authors are then testing the model using test sets that do not follow the same distribution than training data, example,  longer sequences. By doing so, they are breaking one of the most fundamental assumptions of inductive machine learning, i.e., the distribution of train and test data should be equal. Accordingly, my main point is the following: the model is indeed learning the task, as measured by performance on training set, so authors are only showing that the solution selected by the RNN does not follow the one that seems to be used by humans. Importantly, this does not entail that using a better regularization a similar RNN model can indeed learn such a representation. In this sense, the paper would really produce a more significant contribution is the authors can include some ideas about the ingredients of a RNN model, a variant of it, or a different type of model, must have to learn the compositional representation suggested by the authors, that I agree present convenient generalization capabilities.


Anyway, I believe the paper is interesting and the authors are exposing interesting facts that might be worth to spread in our community, so I rate the paper as slightly over the acceptance threshold.

---

> ### Author Response · Authors · 2017-12-07
> **Thanks and please see below**
>
> Thanks for your constructive feedback. We reply to all reviewers jointly in our first comment to the third reviewer below.

---

### Decision · Program_Chairs · 2018-01-29
**ICLR 2018 Conference Acceptance Decision**

**Decision:**

Invite to Workshop Track

**Comment:**

Reviewers were somewhat lukewarm about this paper, which seeks to present an analysis of the limitations of sequence models when it comes to understanding compositionality. Somewhat synthetic experiments show that such models generalise poorly on patterns not attested during training, even if the information required to interpret such patterns is present in the training data when combined with knowledge of the compositional structure of the language. This conclusion seems as unsurprising to me as it does to some of the reviewers, so I would be inclined to agree with the moderate enthusiasm two out of three reviewers have for the paper, and suggest that it be redirected to the workshop track.

Other criticisms found in the review have to do with the lack of any discussion on the topic of how to address these limitations, or what message to take home from these empirical observations. It would be good for the authors to consider how to evaluate their claims against "real" data, to avoid the accusation that the conclusion is trivial from the task set up.

Therefore, while well written, it is not clear that the paper is ready for the main conference. It could potentially generate interesting discussion, so I am happy for it to be invited to the workshop track, or failing that, to suggest that further work on this topic be done before the paper is accepted somewhere.